# Oxycodone-Naloxone Combination Hinders Opioid Consumption in Osteoarthritic Chronic Low Back Pain: A Retrospective Study with Two Years of Follow-Up

**DOI:** 10.3390/ijerph192013354

**Published:** 2022-10-16

**Authors:** Enrico Polati, Marta Nizzero, Jacopo Rama, Alvise Martini, Leonardo Gottin, Katia Donadello, Giovanna Del Balzo, Giustino Varrassi, Franco Marinangeli, Alessandro Vittori, Erica Secchettin, Vittorio Schweiger

**Affiliations:** 1Anesthesiology, Intensive Care and Pain Therapy Centre, Department of Surgery, Dentistry, Paediatrics and Gynaecology, University of Verona, 37124 Verona, Italy; 2Department of Medicine and Public Health, Section of Forensic Medicine, University of Verona, 37124 Verona, Italy; 3Paolo Procacci Foundation, 00193 Rome, Italy; 4Department of Life, Health and Environmental Sciences, University of L’Aquila, 67100 L’Aquila, Italy; 5Department of Anesthesia and Critical Care, ARCO, Ospedale Pediatrico Bambino Gesù IRCCS, 00165 Rome, Italy

**Keywords:** oxycodone, oxycodone-naloxone, opioid tolerance, chronic pain, low back pain

## Abstract

Chronic low back pain (CLBP) due to osteoarthritis represents a therapeutic challenge worldwide. Opioids are extensively used to treat such pain, but the development of tolerance, i.e., less susceptibility to the effects of the opioid, which can result in a need for higher doses to achieve the same analgesic effect, may limit their use. Animal models suggest that ultra-low doses of opioid antagonists combined with opioid agonists can decrease or block the development of opioid tolerance. In this retrospective study, we tested this hypothesis in humans. In 2019, 53 patients suffering from CLBP were treated with either Oxycodone and Naloxone Prolonged Release (27 patients, OXN patients) or Oxycodone Controlled Release (26 patients, OXY patients). The follow-up period lasted 2 years, during which 10 patients discontinued the treatment, 5 out of each group. The remaining 43 patients reached and maintained the targeted pain relief, but at 18 and 24 months, the OXY patients showed a significantly higher oxycodone consumption than OXN patients to reach the same level of pain relief. No cases of respiratory depression or opioid abuse were reported. There were no significant differences in the incidence of adverse effects between the two treatments, except for constipation, more common in OXY patients. From our results, we can affirm that a long-term opioid treatment with oxycodone-naloxone combination, when compared with oxycodone only, may significantly hinder the development of opioid tolerance. We were also able to confirm, in our cohort, the well known positive effect of naloxone in terms of opioid-induced bowel dysfunction incidence reduction.

## 1. Introduction

### 1.1. Opioids for Chronic Pain: Misuse and Opioid Epidemic

Chronic pain is “a pain which persists past the normal time of healing [1]” and is a common problem with a significant socio-economic impact from diagnosis to full recovery [2]. About 30% of adults in the United States and 20% in Europe experienced chronic pain [3,4]. As recommended by the World Health Organization, opioids are the most effective available treatment for chronic cancer pain [5], and in recent years they have also become a mainstay for the treatment of chronic non-cancer pain [6,7,8]. As a natural consequence, we all have witnessed a massive increase in opioid consumption (each year over 4 million people receive a prescription for an opioid), recently leading to the so-called opioid epidemic in the U.S. and prompting the U.S. Dept. of Health and Human Services (HHS) to declare a public health emergency in 2017 [9]. Upsetting rates and numbers were made public from the National Center for Health Statistics (NCHS) in 2020 and 2021, recording over 11 million people misusing prescription opioids in 2019 and 48,006 deaths attributed to overdosing on synthetic opioids other than methadone (in a 12 month period ending June 2020) [10,11]. Moreover, rates of side effects have increased, limiting the clinical utility of these drugs. Common opioid side effects include opioid-induced bowel dysfunction (OIBD), sedation, respiratory depression, euphoria or dysphonia, and itching [12,13].

### 1.2. Opioid-Induced Bowel Dysfunction (OIBD)

Although most opioid side effects subside with chronic use, OIBD mostly persists [14], and it is, therefore, the most frequently reported adverse event in patients receiving opioid treatment [15].

The most common symptom of OIBD is constipation, which occurs in approximately 40% of patients and can negatively affect patients’ quality of life [14,15,16]. The adverse effects of opioids on gastrointestinal function arise from the interaction between opioids and their peripheral receptors in the gastrointestinal tract [17]. It should, therefore, be possible to block the effects of opioids on bowel function through the administration of an opioid antagonist with selective activity on the gut and limited systemic bioavailability. In 2006 was marketed Targin® (Mundipharma, Cambridge, UK), which is a combination tablet of oxycodone and naloxone. Following oral administration, naloxone undergoes extensive first-pass hepatic metabolism and has a low systemic bioavailability of about 2% [18], so it acts almost exclusively on opioid receptors in the gastrointestinal tract, and its central effects are none or minimal [18,19]. Clinical trials have, therefore, demonstrated the ability of oxycodone/naloxone association to prevent and reverse OIBD [20,21] without reducing oxycodone’s intrinsic analgesic efficacy [22,23].

### 1.3. Opioid Tolerance

Another limit of long-term therapy with opioids is the development of tolerance to their analgesic effect. Opioid tolerance is characterized by a reduced susceptibility to the effects of the opioid, which can result in a need for higher and more frequent doses to achieve the same analgesic effect, possibly further limiting the clinical utility of these drugs [23,24]. Although tolerance to the analgesic effects of opioids has been demonstrated in patients in pain, the magnitude of the problem is debated [25,26]. Some studies estimate tolerance incidence at 25% [23]. The pathological mechanisms of tolerance are quite complex and largely unknown and involve the interplay of multiple regulatory mechanisms occurring both at the level of individual opioid-responsive cells and at the level of neuronal circuits [26,27]. Tolerance involves degrees of desensitization and down-regulation, depending on the characteristic of the opioid agonist, although receptor down-regulation is determinant but not necessary for opioid tolerance [28,29]. The mu-opioid receptors’ (MORs) tolerance is not only agonist-dependent, but it also depends on the opioid dose, the route of administration, the duration of the receptors’ exposure and the cellular environment in which receptors are expressed [28,30]. Finally, prolonged opioid treatment activates glial cells in the central nervous system, leading to tolerance development [31,32,33]. Recent studies consider toll-like receptor 4 (TLR4) to play a critical role in mediating opioid-induced glial activation and pro-inflammatory cytokine release [34,35,36]. Opioid antagonists, such as naloxone and naltrexone, used at ultra-low doses in combination with opioid agonists, can decrease or block the development of opioid tolerance in rodents [21,37,38]. Animal models suggest that naloxone has a biphasic dose-dependent effect on pain and that low doses of opioid antagonist could paradoxically produce analgesia [33]. Although clinical experiences with opioid antagonists combined with opioid agonists in humans show that ultra-low doses of naloxone or naltrexone have some analgesic properties and may reduce opioid tolerance [35,36,37,38], there are no specific studies demonstrating that such drugs can reduce the long-term development of opioid tolerance. The hypothesis we tested in this retrospective study is that the combination at fixed doses of an opioid antagonist, such as naloxone, to an opioid agonist, such as oxycodone, can reduce the development of long-term opioid tolerance in humans. To verify the validity of this hypothesis, we evaluated the impact of oxycodone-naloxone combination compared with oxycodone alone on the development of tolerance to the opioid’s analgesic efficacy in patients affected by chronic low back pain due to osteoarthritis.

## 2. Materials and Methods

### 2.1. Study Design and Procedures

In this retrospective study, we collected data from all CLBP patients with a score > 60 according to the Visual Analogue Scale (VAS; 0–100) who visited our pain treatment center at the Borgo Roma Hospital in Verona and started a treatment with orally administered oxycodone-naloxone combination (Targin^®^, Mundipharma, Cambridge, UK) or oxycodone alone (Oxycontin^®^, Mundipharma, Cambridge, UK), from January to December 2019. A rigorous outpatient follow-up appointment schedule was applied, with checkpoints at 0, 1, 3, 6, 12, 18, 24 months from baseline visits. At our pain center, opioid treatment for chronic non-oncological pain is routinely prescribed after compiling the Opioid Risk Tool (ORT), a validated screening instrument commonly and widely used in clinical practice to exclude patients with high or moderate opioid addiction risk [39]. A Registry of Pain was recently developed and introduced in clinical practice once approved by Verona and Rovigo IRB 1751 CESC; details on analgesic assumption and follow-up visits are routinely gathered and recorded for every single case. All patients are asked to sign the written consent form for clinical data management during baseline visit. Moreover, those eligible for opioid treatment for chronic non-cancer pain undergo a titration protocol with morphine sulphate oral solution for seven consecutive days in order to determine the minimal effective daily dose of morphine necessary to maintain a 30% decrease in pain intensity. In case a patient could not reach and be able to maintain the targeted pain relief, the medication dosage was adjusted accordingly by either a pain specialist or general practitioner, and all modifications were recorded. We also kept in regular phone contact with said general practitioners for updates and prescribed the minimum drug dose needed until the subsequent scheduled check-up at our pain center.

The study was carried out according to the local ethical committee (VR-RO Scientific Ethics Committee) guidelines.

### 2.2. Data Collection

We performed a retrospective analysis of data collected from January 2019 to December 2021 and referring to the whole year of 2019. All clinical data were assessed at our department by the physicians during the institutional scheduled medical examination for CLBP patients (1, 3, 6, 12, 18 and 24 months after the start of therapy). The collected clinical data inquired about the VAS score changes from baseline to the standardized subsequent checkpoints, as routinely done in our center. We further analyzed the total daily consumption of opioids and the bowel function using the Bowel Function Index (BFI) to assess opioid-induced constipation [16]. Frequency of treatment-related adverse events (AEs), the incidence of opioid abuse as diagnosed according to DSM-5 criteria [11], and all reasons that led to discontinuation of therapy were also reported.

### 2.3. Statistical Analysis

The statistical analysis was performed comparing data of patients treated with oxycodone only versus those treated with oxycodone and naloxone using non-parametric tests, more robust than parametric. Descriptive statistics were reported in terms of medians and interquartile range for quantitative variables and in terms of absolute frequencies and percentages for qualitative variables. For the comparison of VAS values and drug consumption, we used the Mann–Whitney U test. For demographic data, clinical features, reported opioid side effects, and adverse events, we compared the proportions of patients per group. For the comparison of patients with BFI ≥ 60, we used the chi-square test to evaluate relative frequencies. All tests were two-sided with a significance level set at 5%.

Statistical analysis was performed using MedCalc for Windows, version 11.3.0.0 (MedCalc Software, Ostend, Belgium) and GraphPad Prism^®^ for Windows, version 5.01 (GraphPad^®^ Software, San Diego, CA, USA).

## 3. Results

Data from 53 patients receiving an opioid treatment for at least 2 years and regularly attending the scheduled follow-up visits were extracted: among these, 27 received the drug combination (i.e., Oxycodone and Naloxone Prolonged Release, OXN PR group), and 26 received Oxycodone only (Oxycodone Controlled Release, OXY CR group). The subdivision of data sets into two groups (i.e., OXY and OXN) was made for analysis purposes, only. Data gathering was extended up to 2 years of follow-up time, during which five patients dropped out of the OXN group (two for excessive sedation, one for dizziness, one for stupor and one for gastritis) and five patients dropped out of the OXY group (two for excessive sedation, two for worsening constipation and one for nausea). Final analysis included 43 patients: 21 patients receiving oxycodone only (OXY group) and 22 patients receiving both oxycodone and naloxone (OXN group). The OXY group and the OXN group were similar for demographic data and baseline clinical features (Table 1). Patients taking adjuvant drugs and non-opioid analgesics continued to take them during the study period.

Opioid requirements during the study period are shown in Table 2. In the first year of the study (1, 3, 6, 12 months), the daily consumption of oxycodone PR compared to oxycodone/naloxone PR to reach the targeted pain relief versus baseline did not demonstrate any significant difference between the two groups, while at 18 and 24 months the patients in the OXN group showed a significantly lower oxycodone consumption to reach the same pain relief (Figure 1).

It is important to stress the fact that all 43 patients reached and maintained the targeted pain relief of 30% in terms of VAS reduction versus baseline, as shown in Figure 2. There were no statistically significant differences in the incidence of opioid adverse effects in the two groups except for constipation (Table 3). The number of patients with moderate to severe constipation (BFI ≥ 60) was initially similar in the two groups; however, it trended significantly higher in the OXY group after 18 months of treatment (Figure 3). No patient developed opioid abuse during the 2 years of treatment, and there were no cases of respiratory depression.

## 4. Discussion

This study demonstrated that patients treated with oxycodone-naloxone reached the same pain relief as the ones receiving oxycodone only. However, the patients in the OXN group required lower opioid doses at 18 and 24 months to maintain the same pain relief as those in the OXY group. Moreover, the OXN group had a significantly lower incidence of constipation at 18 and 24 months.

The latter results were already well known in the literature [40,41,42,43,44,45,46,47]. What was not clear was the potential of the oxycodone-naloxone combination to prevent opioid tolerance. Little evidence is available for that matter, and mainly was from animal models or in vitro studies [47,48], mostly controversial or related to other strong opioids [48,49], if not retracted for inconsistencies [50].

The use of opioid antagonists to decrease the development of opioid tolerance as well as their side effects while preserving opioid analgesia is an attractive concept. Recent evidence in rodents indicates that the co-administration of ultralow doses of opioid antagonists with opioid agonists can decrease or block the development of opioid tolerance [21,35,37,38,39]. The exact mechanism of the analgesic effect of ultra-low doses of naloxone is not fully understood; however, some suggestions have arisen. In animal models, naloxone was reported to inhibit microglia activation and superoxide generation and thus protect neurons from injuries [33]. Because neuronal apoptosis is a pathological process occurring in opioid tolerance, the neuroprotective effect of naloxone may inhibit opioid tolerance development. Furthermore, naloxone is also a TLR4 antagonist, and it can be used to attenuate glial activation, alleviating opioid-induced tolerance [34,35,36]. In this study, we observed that the administration of oxycodone-naloxone significantly reduced the development of long-term opioid tolerance and, therefore, the need for increasing opioid doses to obtain and maintain the same level of analgesia, i.e., a 30% reduction in the VAS score (0–100) versus baseline at all checkpoints (Figure 3). In accordance with previous studies [20,21,22,23], our data show that the combination of ultralow doses of naloxone and oxycodone did not negatively affect the analgesic efficacy of oxycodone. In the long term period, this combination enabled us to steadily reach the targeted pain relief with lower opioid doses (Figure 1). Regarding the specific effect of naloxone on the gut to block the effects of opioids on bowel function with a limited systemic bioavailability [14,18], after 1 year of treatment in our study, a higher, though not statistically significant, number of patients in the OXY group compared to the OXN group were suffering from moderate or severe OIBD. Analyzing the time course of moderate or severe OIBD in the two groups, starting from the 18th month of treatment, we found a statistically significant difference in bowel function compared to previous studies performed on a greater sample size [20,40]. As regards the other opioid-related adverse effects, we did not find any significant differences compared to the literature [12,13,14,23].

In this study, we observed that the incidence of therapy dropouts was similar in the OXY and the OXN groups. In the OXN group, no patients dropped out of the treatment schedule due to excessive constipation, in contrast from patients in the OXY group, providing further evidence that the co-administration of naloxone and oxycodone has a minor impact on bowel function also in the long term. Of great interest is that none of the patients developed opioid abuse while following the treatment schedule. In our opinion, this is mainly due to the careful selection of candidates for chronic opioid therapy that we routinely perform in our Pain Relief Center.

This study has several limitations. The most important is the reduced number of patients. This was due to the strict protocol we routinely apply at our center ahead of patient selection for long-term opioid treatment. Moreover, it was not possible to clarify why patients had received either oxycodone-naloxone or oxycodone only, but the number of patients receiving either therapy was similar, providing the possibility to compare the results. A further limitation could be that the sample primarily comprised Caucasian patients. It is possible, though unlikely, that co-treatment of pain with oxycodone and naloxone may have different effects in other ethnicities. Moreover, the additional administration of non-opioid analgesics was not standardized. It is possible that these drugs influenced the pain scores among the two treatment groups. However, none of the patients underwent modifications of the non-opioid analgesic therapy during the period of time covered by the study.

The strengths of this study include a long, standardized follow-up period (2 years), a homogeneous population (age, same pain diagnosis and VAS score range, narrow timing for inclusion) and an evenly distributed rate of dropouts (18% in total, i.e., 10/53) between the two analyzed groups (5/27, i.e., 18% for OXN, 5/26, i.e., 19% for OXY). Importantly, we aimed to achieve feasible and stable pain relief for our patients, and this enabled us to affirm that the addition of naloxone to oxycodone reduced the development of opioid tolerance and the incidence of OIBD in the long term.

In our cohort, the drug consumption over time to reach and maintain the targeted 30% pain relief was significantly different between the two groups, and this significance grew stronger with time (*p*-value 0.0142 at 18 months, *p*-value 0.0052 at 24 months).

## 5. Conclusions

In conclusion, this is the first real-life and long-term study providing evidence that the oxycodone-naloxone combination may inhibit the development of opioid tolerance in chronic low back pain due to osteoarthritis. Further studies with prospective designs and randomization are needed to better clarify the mechanism of this phenomenon.

## Figures and Tables

**Figure 1 ijerph-19-13354-f001:**
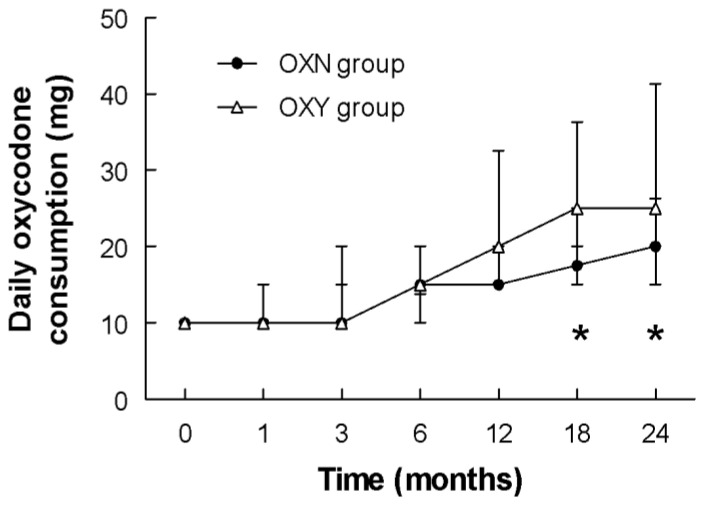
Time course of daily oxycodone consumption (mg) to reach the target 30% pain relief versus baseline. Data are expressed as median and interquartile range (IQR, 25–75). OXY group: oxycodone; OXN group: oxycodone-naloxone. * *p* < 0.05 between the two groups at the different time points.

**Figure 2 ijerph-19-13354-f002:**
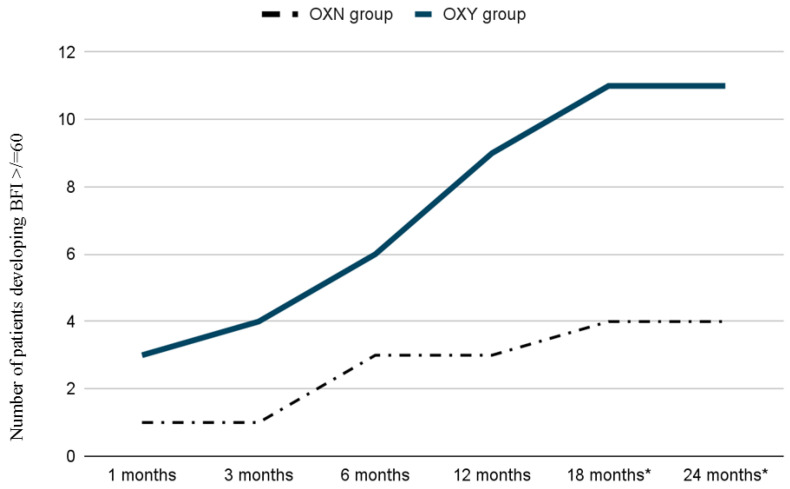
Distribution of patients (absolute number) developing BFI ≥ 60 during follow-up time. OXY group: oxycodone; OXN group: oxycodone-naloxone. * Statistically significant differencereached (*p* < 0.05).

**Figure 3 ijerph-19-13354-f003:**
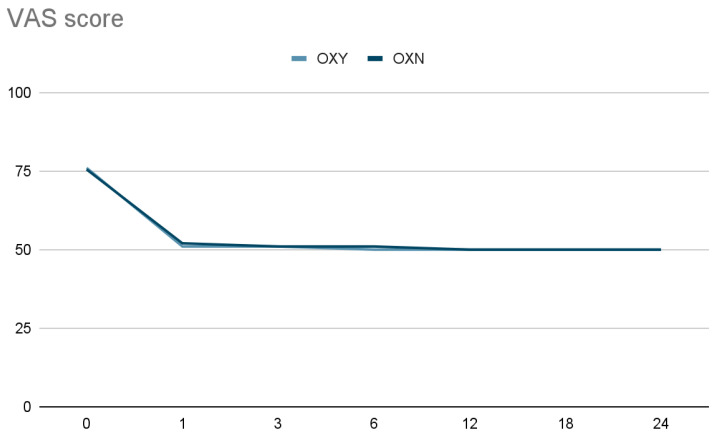
Distribution of VAS values (0–100) expressed in median and IQR (25–75) at the scheduled 2-year follow-up checkpoints (1, 3, 6, 12, 18, 24 months) in the two different study groups (OXY, OXN). All 43 patients reached and maintained the targeted pain relief of 30% versus baseline. OXY group: oxycodone; OXN group: oxycodone-naloxone.

**Table 1 ijerph-19-13354-t001:** Values are expressed as median [IQR, 25–75] for quantitative variables and as number of patients for qualitative variables. Group OXY, oxycodone; Group OXN, oxycodone + naloxone. SSRI, Selective Serotonin Reuptake Inhibitors, SNRI, Selective Noradrenaline Reuptake Inhibitors, TCA Tricyclic Acids. The baseline BFI score difference between the two groups was not statistically significant.

Demographics and Baseline Clinical Features Variables (n = 43)		
	Group OXN	Group OXY
	n = 22	n = 21
Age (years)	74.5 (70.0–78.0)	73.0 (64.8–82.3)
Male	7	7
Female	15	14
Caucasians	22	21
Non Caucasians	0	0
VAS at baseline (0–100)	75.5 (68.0–83.0)	76.0 (65.8–85.3)
BFI at baseline (0–100)	32.5 (22.0–50.0)	37.8 (25.4–51.4)
Start opioid daily dosage (mg)	10.0 (10.0–10.0)	10.0 (10.0–10.0)
Non opioid analgesics (paracetamol, NSAIDs)	18	19
Gabapentinoids at baseline	7	6
SSRI at baseline	4	3
SNRI at baseline	2	3
TCA at baseline	3	2

**Table 2 ijerph-19-13354-t002:** Comparison of drug dosage between two groups across time, at different timepoints (baseline, 1, 3, 6, 12, 18 and 24 months). Values are expressed as median [IQR 25–75]. Group OXY, oxycodone; Group OXN, oxycodone + naloxone.

Daily Oxycodone PR Consumption in mg							
Group	Time (months)						
	0	1	3	6	12	18	24
OXY	10.0 (10.0–10.0)	10.0 (10.0–10.0)	10.0 (10.0–15.0)	15.0 (13.8–20.0)	20.0 (15.0–32.5)	25.0 (20.0–36.3)	25.0 (20.0–41.3)
OXN	10.0 (10.0–10.0)	10.0 (10.0–15.0)	10.0 (10.0–20.0)	15.0 (10.0–20.0)	15.0 (15.0–20.0)	17.5 (15.0–25.0)	20.0 (15.0–26.3)
*p*	0.8181	0.4146	0.7503	0.1326	0.1067	0.0142	0.0052

**Table 3 ijerph-19-13354-t003:** Reported opioid side effects or adverse events as recorded per single group (OXY vs. OXN). Values are expressed as the number of patients. Group OXY, oxycodone; Group OXN, oxycodone + naloxone.

Reported Opioid Side Effects or Adverse Events		
	Group OXN	Group OXY
	n = 27	n = 26
Nausea	4	3
Vomiting	0	0
Lightheadedness	2	2
Myoclonus	1	0
Itching	1	0
Moderate to severe constipation (BFI ≥ 60)	4	11 *
Drowsiness/sedation	5	6
Strong sedation	2	2
Dizziness	1	1
Dry mouth	0	0
Fatigue	0	1
Cognitive impairment	0	0
Hyperalgesia/allodynia	0	0
Respiratory depression	0	0
Gastritis	1	0
Stupor	1	0

* *p* < 0.05 vs. other group.

## Data Availability

Not applicable.

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
