# Peer review of "Oxycodone-Naloxone Combination Hinders Opioid Consumption in Osteoarthritic Chronic Low Back Pain: A Retrospective Study with Two Years of Follow-Up"

_ijerph, 2022, doi:10.3390/ijerph192013354_

Round 1
Reviewer 1 Report
The presented Article, Oxycodone- Naloxone combination hinders the development of opioid tolerance in chronic non cancer pain: a retrospective study with two years of follow-up by Polati E. et al. compares the effectiveness of treatment of two opioid analgesics. The authors applied a 2-year-lasting therapy with oxycodone alone (OXY) or oxycodone combined with naloxone (OXN) in patients with chronic low back pain due to osteoarthritis. They analysed the opioid tolerance development and side effects occurrence during applied therapy.
Generally, the arrangement and design of this study are not rigorous enough, and there are many major issues which should be addressed to see the novelty of undertaken studies as well as to see whether the conclusion about lower risk of opioid tolerance in case of long term OXN therapy vs long term OXY therapy is valid. My critical remarks concern the study's aim, introduction, methodological elucidations, results presentation, result description, and discussion of obtained results. The manuscript should be thoroughly corrected and completed; therefore, I do not recommend publishing the manuscript in the publication in International Journal of Environmental Research and Public Health.
Specific Comments:
1. The title does not reflect the Article content. Inconsistences I found in the title vs body text of the Article:
a)The title says about opioid tolerance, while in the manuscript, no data shows the level of opioid tolerance in studied groups of patients
b)Based on the body text, authors performed studies in specific "non cancer pain" groups of patients. The phrase "non cancer pain" is not precise because analysed patients suffered only one kind of not cancer patients - chronic low back pain due to osteoarthritis.
b)in the Article authors did not convince readers that the type of presented studies is retrospective, prospective or other. Mainly because the sample size is smaller than usual in retrospective studies; there is no information if/how authors prevented selection bias between groups. Therefore the title seems to be overstated
Please correct the title of the manuscript and if applicable, tone it down.
2. The sentence in lines 22-23: "Animal models...." is confusing because oxycodone-naloxone therapy is also well-known as an effective pain therapy in clinics. See, e.g. 1) Pain Pract. 2016 Jun;16(5):580-99. DOI: 10.1111/papr.12308. Epub 2015 Jun 12. Please rephrase to indicate the rationale for undertaken studies.
3. The introduction contains information hardly related to the manuscript's primary goal, which is the comparison of OXN to OXY treatment applied for patients with "Chronic low back pain (CLBP) due to osteoarthritis". Please correct the introduction and put information specifically related to the described problem here.
4. Please explain all abbreviations used in the Article, e.g. HHS (line 45); NCHS (line 46); IQR (lines 153, 164, 234). Additionally, improve consistency with abbreviations usage. Did you mean CLBP (line 20) or LBP (line 117)?
5. Study design and procedures are not described precisely. I recommend completing, in the body text of the Material and Methods section, patients' age, comorbidities, the method of patient selection into groups, method of patient observation etc.
Moreover, please describe the way of spondyloartrosis diagnosis in patients who served as a model for the therapy comparison in your study. VAS can be used to measure pain intensity. However, it is not explicitly dedicated to "LBP due to spondyloarthritis" (lines 101-102). Additionally, please complete information about the degree of spine degeneration of patients in your studies, and give more detailed information than "≥18 yrs." information about the age of patients.
6. Based on comment #5, please elaborate on the influence of age, and other disease factors on diagnosed pain in patients and, if applicable, adjust (tone down) your conclusions, and the title of the Article for observed limitations (see also comment #1)
7. In lines 103 and 116, the phrase "our department" is not precise. Please add details for a place of performed study with clear information on whether patients were hospitalised or under clinical care. If patients were not hospitalised, please add information about the schedule of appointments. Explain also how to discipline for taking only prescription medications was controlled.
8. There is no criterion of patient recruitment for "oxycodone-naloxone" and "oxycodone alone" treatment. Please complete.
9. Please add a number of appropriate ethical committee approval for conducting a study on people.
10. line 115: the phrase "We performed a retrospective analysis of prospective data" is confusing. Please rephrase or state and define precisely how you performed retrospective and prospective studies.
11. The content of table 2 is not straightforward. Please add rationale (in the manuscript body text) of performed statistics comparing daily oxycodone consumption in OXY and OXN groups. Additionally, please explain the numbers in brackets.
12. In the Result section, the data "demonstrating that the oxycodone-naloxone combination inhibits the development of opioid tolerance in chronic non cancer pain" is missing. Please complete or explain better how you measured the development of tolerance.
13. Please provide appropriate (consistent with information in lines 125-136) statistical reports for all data you analysed and also for data where you did not find significance.
14. Discussion is sparse regarding of meaning of obtained data and needs correction.
My main criticism of the current version is because of the lack of confronting of obtained data with well-known literature data about the advantages of combining naloxone with oxycodone in one drug.
Example 1. Better analgesic efficacy of OXN with lower risk of gastrointestinal side effects, especially constipation, is known from clinical studies to see, e.g. Eur J Pain; Int J Clin Pract
. 2008 Aug;62(8):1159-67. DOI: 10.1111/j.1742-1241.2008.01820.x.
. 2009 Jan;13(1):56-64. DOI: 10.1016/j.ejpain.2008.06.012.
example 2: Authors state, " the mechanism of improved safety of opioid treatments by combination naloxone with oxycodone is at least partially described in the literature.
Please elaborate on the discussion section with a description of the meaning of the presented results based on the current literature data.
15. Please rephrase and give merit arguments for statements like "careful selection" (line 209)
16. line 226-229. The presented data does not support conclusions (no data showing tolerance development). Please improve consistency.
Author Response
Comment 1: The title does not reflect the Article content. Inconsistences I found in the title vs body text of the Article:
a)The title says about opioid tolerance, while in the manuscript, no data shows the level of opioid tolerance in studied groups of patients
b)Based on the body text, authors performed studies in specific "non cancer pain" groups of patients. The phrase "non cancer pain" is not precise because analysed patients suffered only one kind of not cancer patients - chronic low back pain due to osteoarthritis.
c)in the Article authors did not convince readers that the type of presented studies is retrospective, prospective or other. Mainly because the sample size is smaller than usual in retrospective studies; there is no information if/how authors prevented selection bias between groups. Therefore the title seems to be overstated
Please correct the title of the manuscript and if applicable, tone it down.
Response 1a: The aim of the present study was not to inquire about opioid tolerance itself, rather more on the novelty of naloxone potential in preventing the development of opioid tolerance at very low dosage. Nevertheless, as reported in table 2 and inherent text we discussed effects on opioids tolerance as thoroughly as it was needed according to our main objectives. We proceeded to tone the title down, anyways.
Response 1b: We modified the text accordingly, referring to the sole entity of chronic low back pain due to ostheoarthritis.
Response 1c: We modified the text accordingly and specified the exact timing of data gathering with respect to the time when follow-ups were scheduled (i.e. data gathering occurring from January 2019 to December 2021 and baseline visits occurring during year 2019, January to December). The sample is relatively small, but includes the total amount of patients who referred to our centre for CLBP due to osteosarthritis and did not drop out of the treatment schedule during the two-years long follow-up period. No patient selection was carried out, the choice of starting a patient on either oxycodone alone or oxycodone and naloxone combined was totally on the physician depending on clinical features, comorbidities, likelihood to develop OIBD or tolerance, and was made at the time of baseline.
Comment 2. The sentence in lines 22-23: "Animal models...." is confusing because oxycodone-naloxone therapy is also well-known as an effective pain therapy in clinics. See, e.g. 1) Pain Pract. 2016 Jun;16(5):580-99. DOI: 10.1111/papr.12308. Epub 2015 Jun 12. Please rephrase to indicate the rationale for undertaken studies.
Response 2: The aim of the present study was to inquire on naloxone role in preventing the development of opioid tolerance. We are well aware of the rich evidence provided by scientific literature on the analgesic effectiveness of oxycodone-naloxone combination, see also
- Kim, E.S. Oxycodone/Naloxone Prolonged Release: A Review in Severe Chronic Pain. Clin Drug Investig 37, 1191–1201 (2017)
- Burness, C.B., Keating, G.M. Oxycodone/Naloxone Prolonged-Release: A Review of Its Use in the Management of Chronic Pain While Counteracting Opioid-Induced Constipation. Drugs 74, 353–375 (2014)
- Poelaert et al, Treatment With Prolonged-Release Oxycodone/Naloxone Improves Pain Relief and Opioid-Induced Constipation Compared With Prolonged-Release Oxycodone in Patients With Chronic Severe Pain and Laxative-Refractory Constipation, Clin Ther, 37, Issue 4, 784-792 (2015)
What we wanted to inspect was, instead, the role of low doses of naloxone in tolerance prevention. Little evidence is available for that matter, and mainly animal models or in vitro studies, mostly controversial (see "Wang et al, High-Affinity Naloxone Binding to Filamin A Prevents Mu Opioid Receptor–Gs Coupling Underlying Opioid Tolerance and Dependence. Published: February 6, 2008" which was repeatedly retracted for inconsistencies).
Comment 3. The introduction contains information hardly related to the manuscript's primary goal, which is the comparison of OXN to OXY treatment applied for patients with "Chronic low back pain (CLBP) due to osteoarthritis". Please correct the introduction and put information specifically related to the described problem here.
Response 3: As previously clearified, the main objective of the study was not to inquire on the comparison between OXN and OXY treatment of CLBP due to ostheoarthritis. We proceeded to better specify, in the discussion section, that our main goal was to examine whether low doses of naloxone could have an impact on the development of tolerance and/or analgesic effect of the combination drug.
Comment 4. Please explain all abbreviations used in the Article, e.g. HHS (line 45); NCHS (line 46); IQR (lines 153, 164, 234). Additionally, improve consistency with abbreviations usage. Did you mean CLBP (line 20) or LBP (line 117)?
Response 4: All abbreviations were explained correctly. The text has been modified using the acronym CLBP (vs LBP) only.
Comment 5.
Part I. Study design and procedures are not described precisely. I recommend completing, in the body text of the Material and Methods section, patients' age, comorbidities, the method of patient selection into groups, method of patient observation etc.
Part II. Moreover, please describe the way of spondyloartrosis diagnosis in patients who served as a model for the therapy comparison in your study. VAS can be used to measure pain intensity. However, it is not explicitly dedicated to "LBP due to spondyloarthritis" (lines 101-102). Additionally, please complete information about the degree of spine degeneration of patients in your studies, and give more detailed information than "≥18 yrs." information about the age of patients.
Response 5.
Part I. We modified the contents so that the nature of the study (as in its design and objectives) would be clearer. No specific method of selection was applied since no actual patient selection into groups was made. The drug- therefore treatment group OXY vs OXN- choice was made by the physician during the baseline visit.
Part II. We made clear the VAS scoring of pain was solely referred to CLBP due to osteoarthritis.
Comment 6. Based on comment #5, please elaborate on the influence of age, and other disease factors on diagnosed pain in patients and, if applicable, adjust (tone down) your conclusions, and the title of the Article for observed limitations (see also comment #1)
Response 6: We modified the text accordingly toning both title and conclusions down.
Comment 7:
Part I: In lines 103 and 116, the phrase "our department" is not precise. Please add details for a place of performed study with clear information on whether patients were hospitalized or under clinical care. If patients were not hospitalized, please add information about the schedule of appointments.
Part II: Explain also how discipline for taking only prescription medications was controlled.
Response 7
Part I: We clarified the first part as follows: "(...) who visited our pain treatment centre at the Borgo Roma Hospital in Verona department and started a treatment with orally administered oxycodone-naloxone combination (Targin®, Mundipharma, Cambridge, UK) or oxycodone alone (Oxycontin®, Mundipharma, Cambridge, UK), from January to December 2019. An outpatient rigorous follow-up appointments schedule was carried out, checkpoints being at 0, 1, 3, 6, 12, 18, 24 months from baseline visits.
Part II: As to how to discipline for taking only prescription medications, apart from a strict follow-up schedule, we relied on the fact that opioids can only be prescribed by pain specialists or general practitioners, and it is not possible to obtain said medications without a specific medical prescription. Furthermore, in case a patient would not reach and be able to maintain at least 30% pain relief, the medication dosage was adjusted accordingly and all modifications were recorded. We also kept in regular phone contact with said general practitioners for updates and prescribed the minimum drug dose needed until the subsequent scheduled check-up at our pain center.
Comment 8: There is no criterion of patient recruitment for "oxycodone-naloxone" and "oxycodone alone" treatment. Please complete.
Response 8: No criterion was used since in our retrospective analysis data of all the patients suffering from > 60 VAS score CLBP due to osteoarthritis who did not drop out of treatment schedule were gathered.
Comment 9: Please add a number of appropriate ethical committee approval for conducting a study on people.
Response 9: We did not ask for ethical committee approval since it was a retrospective study. We still followed the VR-RO Ethical Committee guidelines for privacy, data gathering and analysis. The name of the Ethical Committee we referred to for instructions has been added in the manuscript.
Comment 10. line 115: the phrase "We performed a retrospective analysis of prospective data" is confusing. Please rephrase or state and define precisely how you performed retrospective and prospective studies.
Response 10: The text was modified accordingly.
Comment 11. The content of table 2 is not straightforward. Please add rationale (in the manuscript body text) of performed statistics comparing daily oxycodone consumption in OXY and OXN groups. Additionally, please explain the numbers in brackets.
Response 11. We believe that the rationale of fig 1 and table 2 stands in the effectiveness and entirety of information provided by a tabular display and through quantitative statistical indices such as medians and IQR.
Comment 12: In the Result section, the data "demonstrating that the oxycodone-naloxone combination inhibits the development of opioid tolerance in chronic non cancer pain" is missing. Please complete or explain better how you measured the development of tolerance.
Response 12: We modified the discussion section accordingly.
Comment 13. Please provide appropriate (consistent with information in lines 125-136) statistical reports for all data you analysed and also for data where you did not find significance.
Response 13. We modified the text so the rationale and details of performed statistics would be clearer to the reader.
Comment 14. Discussion is sparse regarding of meaning of obtained data and needs correction.
My main criticism of the current version is because of the lack of confronting of obtained data with well-known literature data about the advantages of combining naloxone with oxycodone in one drug.
Example 1. Better analgesic efficacy of OXN with lower risk of gastrointestinal side effects, especially constipation, is known from clinical studies to see, e.g. Eur J Pain; Int J Clin Pract
- 2008 Aug;62(8):1159-67. DOI: 10.1111/j.1742-1241.2008.01820.x.
- 2009 Jan;13(1):56-64. DOI: 10.1016/j.ejpain.2008.06.012.
example 2: Authors state, " the mechanism of improved safety of opioid treatments by combination naloxone with oxycodone is at least partially described in the literature.
Please elaborate on the discussion section with a description of the meaning of the presented results based on the current literature data.
Response 14: We elaborated the discussion section thoroughly, completed and better related literature to our considerations as requested. References were reviewed, updated, modified or supplemented where needed.
Comment 15. Please rephrase and give merit arguments for statements like "careful selection" (line 209)
Response 15: We modified the "methods" section accordingly.
Comment 16. line 226-229. The presented data does not support conclusions (no data showing tolerance development). Please improve consistency.
Response 16. We proceeded to improve consistency adding to the main texts results inferable from tables and figures.
Reviewer 2 Report
Thank you for inviting me to review this manuscript entitled ”Oxycodone- Naloxone combination hinders the development of opioid tolerance in chronic non cancer pain: a retrospective study with two years of follow-up”. The paper itself is well written and provides an interesting observation on the long-term opioid treatment with oxycodone-naloxone combination. The authors have conducted a thorough literature review, undertaken a rigorous piece of data collection and analyzed information accurately. They listed the strengths and weaknesses of their study, with which I agree. I am waiting for more detailed results that clarify the mechanism.
The paper can be accepted, but there are still some changes that you need to make before your paper can be presented.
Minor changes:
-Line 45: Please, add a space in ”[…] 2017[6].”
-The number of the local ethical committee approval.
Author Response
Point 1: Line 45: Please, add a space in ”[…] 2017[6].”
Response 1: The text was corrected accordingly.
-The number of the local ethical committee approval.
Response 2: The local ethical committee approval was not needed as it was a retrospective study. We proceeded with data gathering and with the shaping of the study and its design as suggested by the local ethics committee guidelines. We added specifics on the regional jurisdiction of the committee we referred to for instructions (VR-RO Committee).
Reviewer 3 Report
Introduction: The content presents the problem well, it needs to be separated into more paragraphs.
Materials/Methods: Line 103, "who were visited in our" should be "who visited our"
Discussion: Line 178, "development of opioids tolerance and their side effects" should be "development of opioid tolerance and its related side effects" Line 219, I believe that the conclusion that their is "fair evidence" that the addition of naloxone to oxycodone in chronic low back pain patients reduces the development of opioid tolerance should be a measured conclusion. It may reduce the development of opioid tolerance - without a more diverse, larger sample, and randomization, the authors can say for certain.
Author Response
Comment 1: Materials/Methods: Line 103, "who were visited in our" should be "who visited our"
Response 1: The text was modified accordingly.
Comment 2: Discussion: Line 178, "development of opioids tolerance and their side effects" should be "development of opioid tolerance and its related side effects"
Response 2: The text was modified accordingly.
Comment 3: Discussion: Line 219, I believe that the conclusion that their is "fair evidence" that the addition of naloxone to oxycodone in chronic low back pain patients reduces the development of opioid tolerance should be a measured conclusion. It may reduce the development of opioid tolerance - without a more diverse, larger sample, and randomization, the authors can say for certain.
Response 3: We reelaborated the text as follows "Nevertheless, we "may still assume" (instead of "provide fair evidence") that the addition of naloxone to oxycodone reduces the development of opioid tolerance and the incidence of OIBD in the long term." We subsequently provided evidence as in terms of opioid (mg of oxycodone) consumption along time as follows "Regarding opioid tolerance reduction in fact, we provided evidence that in our cohort the drug consumption (mg) along time to reach and maintain the targeted 30% pain relief was significantly different between the two groups, i.e. OXY and OXN, and that this significance grew stronger with time (p-value 0.0142 at 18 months, p-value 0.0052 at 24 months)."
Round 2
Reviewer 1 Report
The authors improved the manuscript but still, in the reviewer's opinion, the manuscript content does not support the aim of the study.
The authors say that the study aimed "to examine whether low doses of naloxone could have an impact on the development of tolerance and/or analgesic effect of the combination drug". However, there are no convincing data showing the beneficial effect of OXN on tolerance development. Please clarify, in the body text, why you think, based on your results, that taking 2 times more oxycodone per day after 2 years treatment vs the dosage during the first 3 months of therapy with OXN reflects less tolerance than after OXY treatment. Please add to the manuscript data which evidence the stable pain relief in analysed time for all groups. Additionally, please specify the aim of the study in the abstract and the introduction sections
The statement justifying the choice of the statistical method of analysis that non-parametric tests are "more robust than parametric" is not clear. Please explain or correct if necessary by providing reports of parametric statistical analysis.
There is the lack in the manuscript discussion of the obtained effect on constipation increase after OXY treatment which is significantly different than in patients after OXN therapy. Please discuss this effect in the discussion section.
Based on aboved comments and comments in my previous review report the conclusions are not supported by the results and should be adjusted to manuscript content.
Author Response
Comment 1:
1a) The authors say that the study aimed "to examine whether low doses of naloxone could have an impact on the development of tolerance and/or analgesic effect of the combination drug". However, there are no convincing data showing the beneficial effect of OXN on tolerance development.
1b) Please clarify, in the body text, why you think, based on your results, that taking 2 times more oxycodone per day after 2 years treatment vs the dosage during the first 3 months of therapy with OXN reflects less tolerance than after OXY treatment.
1c) Please add to the manuscript data which evidence the stable pain relief in analysed time for all groups.
1d) Additionally, please specify the aim of the study in the abstract and the introduction sections.
Response 1:
1a) We explicited our results in a more readily understandable and exploitable way, both with content and form editing.
1b) According to the definition of opioid tolerance (see Martyn, JA Jeevendra, Jianren Mao, and Edward A. Bittner. "Opioid tolerance in critical illness." New England Journal of Medicine 380.4 (2019): 365-378), which only requires a stably higher (vs control, i.e. pt who did not develop tolerance) opioid dosage assumption to reach and maintain the same pain relief. No minimum increase in terms of dosage assumption is implied in the conventional definition of opioid tolerance.
1c) A new graph was added to display VAS scores distribution along time in the two study groups.
1d) We better clarified the aim of the study modifying the text accordingly.
Comment 2: The statement justifying the choice of the statistical method of analysis that non-parametric tests are "more robust than parametric" is not clear. Please explain or correct if necessary by providing reports of parametric statistical analysis.
Response 2: Based on our population's numerosity and the type of distribution, a non parametric test was deemed more appropriate.
See for completeness:
- for test selection in general, parametric vs non parametric: Mishra P, Pandey CM, Singh U, Keshri A, Sabaretnam M. Selection of appropriate statistical methods for data analysis. Ann Card Anaesth. 2019 Jul-Sep;22(3):297-301.
- for robustness: Usman, Mohammed. Robustness of Parametric and Nonparametric Tests under Non-normality for Two Independent Sample ICAPM, Vol 4, 2018
Comment 3: There is the lack in the manuscript discussion of the obtained effect on constipation increase after OXY treatment which is significantly different than in patients after OXN therapy. Please discuss this effect in the discussion section.
Response 3: We modified the text accordingly.
Text editing was digitally highlighted. Color code: grey for first editing, yellow for second (current) editing.